# Utilization of Response Surface Methodology in Optimization and Modelling of a Microbial Electrolysis Cell for Wastewater Treatment Using Box–Behnken Design Method

**Nhlanganiso Ivan Madondo** [1,*], **Sudesh Rathilal** [1] **and Babatunde Femi Bakare** [2]

1   Green Engineering Research Group, Department of Chemical Engineering, Faculty of Engineering and the Built Environment, Durban University of Technology, Steve Biko Campus, S4 Level 1, Durban 4000, South Africa
2   Environmental Pollution and Remediation Research Group, Department of Chemical Engineering, Mangosuthu University of Technology, Durban 4026, South Africa
*   Correspondence: 21450515@dut4life.ac.za

**Abstract:** A vast quantity of untreated wastewater is discharged into the environment, resulting in contamination of receiving waters. A microbial electrolysis cell (MEC) is a promising bioelectro-chemical system (BES) for wastewater treatment and energy production. However, poor design and control of MEC variables may lead to inhibition in the system. This study explored the utilization of Response Surface Methodology (RSM) on the synergistic aspects of MEC and magnetite nanoparticles for wastewater treatment. Influences of temperature (25–35 °C), voltage supply (0.3–1.3 V) and magnetite nanoparticle dosage (0.1–1.0 g) on the biochemical methane potentials (BMPs) were investigated with the aim of optimizing biogas yield, chemical oxygen demand removal and current density. The analysis of variance (ANOVA) technique verified that the quadratic models obtained were substantial, with *p*-values below 0.05 and high regression coefficients ($R^2$). The optimum biogas yield of 563.02 mL/g $VS_{fed}$, chemical oxygen demand (COD) removal of 97.52%, and current density of 26.05 mA/m$^2$ were obtained at 32.2 °C, 0.77 V and 0.53 g. The RSM revealed a good comparison between the predicted and actual responses. This study revealed the effective utilization of statistical modeling and optimization to improve the performance of the MEC to achieve a sustainable and eco-friendly situation.

**Keywords:** microbial electrolysis cell; response surface methodology; temperature; voltage supply; magnetite; nanoparticle; Box–Behnken design





## 1. Introduction

Sustaining sanitary and fresh water is a major concern for the whole world, therefore the treatment of wastewater for reuse is equally significant [1]. Wastewater usually contains toxic contaminants, which, if not treated when discharged into waterbodies, would lead to water pollution and consequently spread waterborne illnesses. Thus, there is a need for the development of an efficient technique for wastewater purification and treatment before it is discharged into the receiving waterbodies.

Lately, a recent technique that employs the combination of an electromagnetic field and anaerobic digestion has been proven to enhance the activity of microorganisms and remove toxic contaminants [2]. A bioelectrochemical system (BES) is the most promising electromagnetic field method, where exoelectrogens or electroactive microbes enhance the redox mechanisms at the electrode surface. The BES uses electrodes to promote the degradation of organic substrates in anaerobic digestion [3]. Presently, the BES technique has gained more attention in anaerobic digestion, since the use of these two methods generate methane [4–6]. A bioelectrochemical system, also termed a microbial fuel cell (MFC), occurs when electrical voltage is generated from an oxidized biochemical substrate,

and a microbial electrolysis cell (MEC) occurs when voltage is applied to enhance the microbial activity [7]. Amongst BESs, the MEC has been found to be the best-performing system in anaerobic digestion [5]. A MEC process consists of an anodic compartment where organic matter is oxidized, thereby producing protons, electrons, and carbon dioxide. The electrons flow via an external electrical system using electric fields to the cathode compartment, whereas protons and ions flow across a membrane. Protons, electrons, and carbon dioxide react at the cathode compartment to produce methane [8]. The reaction is enhanced by electroactive and hydrogenotrophic microbes.

The MEC is governed by a number of input variables or process control indicators, such as temperature, pH, voltage supply, electrode type, electrode configuration, distance between the electrodes, electrical conductivity, and ionic strength [9,10]. These process control indicators are extremely significant in evaluating how the MEC process functions; they provide an indication of process disturbances that might occur so that preventive actions may be implemented as early as possible. The process control indicators influencing the MEC system have to be correctly controlled and properly designed in order to optimize the efficiency of the process, improve the process stability, and prevent inhibition.

Optimization of process control indicators plays an important part in the treatment of biological wastewater [11,12]. The optimization and relationship of the process control variables can help improve the efficiency of a digester. For this reason, the design of experiments is a good choice for the optimization of the process control variables. Compared to the traditional one-factor-at-a-time method, the design of experiments attains a smaller number of experimental runs and is more efficient and reliable [13]. One of the most powerful techniques that is employed in the design of experiments is response surface methodology (RSM), which is useful for improving the efficiencies of digesters and also examining the effect of several process control variables [14]. The RSM is a combination of mathematical and statistical techniques that can examine modeling to fully understand the correlation of several control variables on the process responses. The quantitative information acquired from the design of experiments and analysis of the RSM model, together with process control variables, can lead to high-end performance [15].

A number of researchers have investigated the utilization of RSM on BESs, with many investigators focusing on optimizing the system. Hosseinpour et al. [16] studied the effect of pH, ionic strength and buffer concentration on power density and COD removal while employing RSM on a microbial fuel cell. The highest power density (32.5 mW/m$^3$) and COD removal (92.5%) were found at the optimum pH (6.75), ionic strength (4.69 mM), and buffer concentration (0.177 M). Power density improved by 17% and COD removal improved by 5% when compared with the control (pH = 7.0, ionic strength = 2.5 mM, and buffer concentration = 0.1 M). The optimization of a microbial fuel cell for enhanced power density and COD removal was investigated by Sedighi et al. [17] using RSM together with central composite design. The focus of the study was on optimizing platinum (0.1 to 0.5 mg/cm$^2$), the degree of sulphonation in SPEEK (20 to 80%), and the rate of cathodic aeration (10 to 150 mL/min) in order to optimize the response variables. Results of the RSM revealed the optimum power density of 58.19 mW/m$^2$ and COD removal of 94.8%. Choi et al. [18] investigated the influence of applied voltage, substrate concentration (food waste), and reactor volume to electrode area ratio on methane production while using the Taguchi technique and RSM. The results of the Taguchi technique and RSM revealed the optimum voltage supply (1.2 V), substrate concentration (2.4 g COD/L), and reactor volume to electrode area ratio (0.33 m$^3$/m$^2$). Regardless of such results in RSM, the optimization of combinations are possibly the most important operating conditions: temperature, voltage supply and magnetite nanoparticle dosage have never been examined in terms of the synergistic aspects of bioelectrochemical systems with magnetite nanoparticles while using the Box–Behnken design method. Moreover, these operating conditions have never been optimized for biogas yield, chemical oxygen demand removal and current density by means of RSM. In anaerobic digestion, magnetite nanoparticles can help to enhance the interspecies electron transfer between archaea and microorganisms. The use of magnetite nanoparticles

together with BESs has proven to enhance electrochemical efficiencies (current density, coulombic efficiency, and electrical conductivity), improve biogas production, and remove more wastewater contaminants (total suspended solids, COD, total organic carbon, color, ammonia nitrogen, and phosphate) [2,5,19].

Thus, this investigation employed RSM optimization and modeling to study the correlation between input variables and responses by determining the predicted model equations. A Box–Behnken design method via RSM was utilized to investigate the influence of three input variables, namely temperature, voltage supply, and magnetite nanoparticle dosage on biogas yield, chemical oxygen demand removed, and current density.

## 2. Results and Discussions

### 2.1. Influence of Input Variables on the Responses

A scatterplot graph is a significant plot in statistical analysis as it measures the effect of each process variable on the output response. With each process variable comprising three coded values (−1, 0, 1), a response versus process variable graph was plotted for each variable (Figure 1). The relationship between the process variable and the response was first assumed to be a linear model. Thus, the gradient of the plots may be used to represent the correlation. A correlation with a positive value signifies that a direct proportionality exists between the process variable and the response output, whereas a correlation with a negative value indicates that the process variable is indirectly proportional to the response output. The correlations (displayed on the top left corner of the graphs) for magnetite nanoparticle dosage, temperature, and voltage supply were −0.100 (Figure 1a), 0.215 (Figure 1b), and −0.055 (Figure 1c), respectively. Therefore, increasing either magnetite nanoparticle dosage or voltage supply resulted in a decrease in biogas yield, whereas a rise in temperature was followed by a rise in biogas yield.

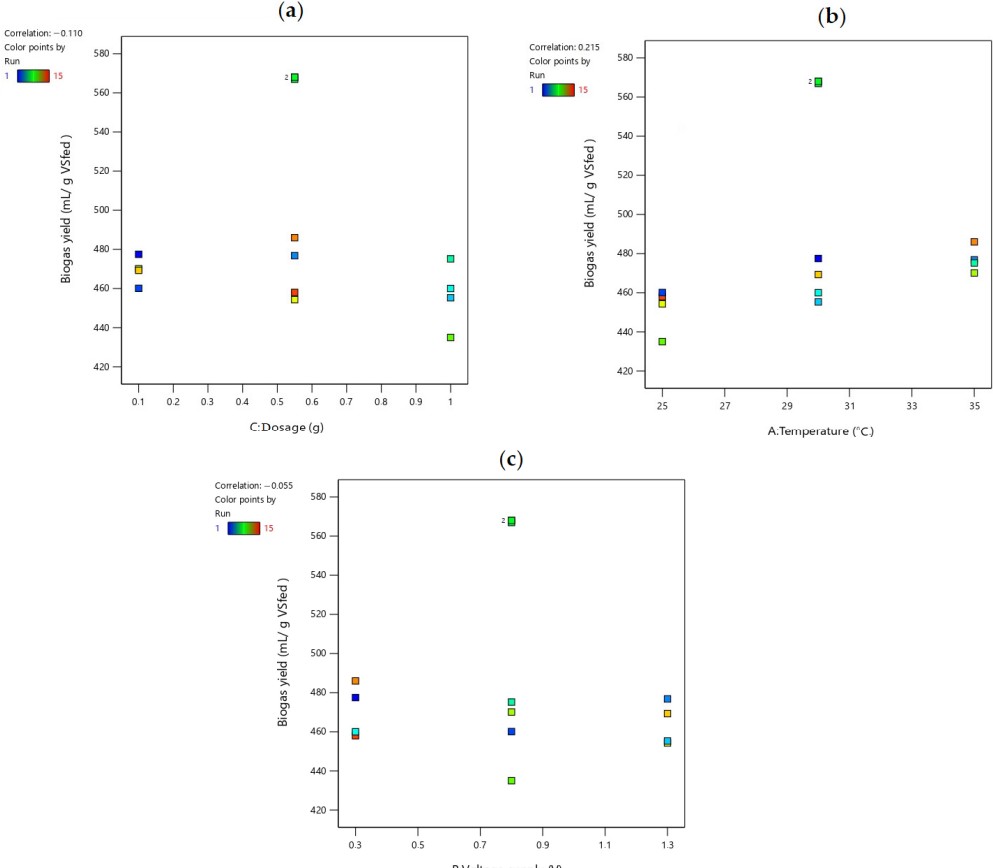

**Figure 1.** Effect of (**a**) magnetite nanoparticle dosage; (**b**) temperature; and (**c**) voltage supply on biogas yield.

The increasing order of the absolute values of the correlations revealed the following: temperature (0.215) > magnetite nanoparticles dosage (0.100) > voltage supply (0.055). Therefore, these results indicated that temperature had the highest effect on biogas yield, whereas voltage supply had the least effect on biogas yield. Similarly, %COD removed and current density were mostly influenced by temperature; their respective correlations (not displayed on the figures) were 0.324 and 0.243 for temperature, −0.222 and −0.177 for magnetite nanoparticle dosage, and −0.148 and −0.045 for voltage supply, respectively.

These results offer a starting point for statistical analysis. The assumption that was made in this section was that the data points resemble a linear model, however, this had to be verified using the fit summary analysis.

### 2.2. Regression Model and Fit Summary

Response transformation is a significant tool of any data analysis. Response power transformation in statistical analysis encompasses the use of several mathematical functions to the response/s. The mathematical functions that were investigated by the Design Expert software for transformation were: no transformation, natural log, inverse square root, square root, base 10 log, logit, inverse, arcsine square root, and power. The transformations for biogas yield, COD removal, and current density models are shown in Table 1. The power transform depends on the maximum response/minimum response ratio [20]. A maximum response/minimum response ratio above 10 normally indicates that the chosen mathematical function must be transformed. On the other hand, a ratio below 3 indicates that the transformation has a small effect [20]. The minimum responses for biogas yield, %COD removed, and current density were 435.0 mL/g $VS_{fed}$, 53.0%, and 12.0 mA/m$^2$, whereas the maximum responses were 568.0 mL/g $VS_{fed}$, 98.0%, and 26.8 mA/m$^2$, respectively. Thus, the maximum response/minimum response ratios for biogas yield, COD removal, and current density were 1.31, 1.85, and 2.23, respectively. For all models, the maximum response/minimum response ratios were below 3, and this indicates that no transformation was required and that transforming the output variables would not make much difference.

**Table 1.** Transformation for biogas yield, %COD removed, and current density models.

| Parameter | Biogas Yield (mL/g VS$_{fed}$) | %COD Removed | Current Density (mA/m$^2$) |
|---|---|---|---|
| Minimum response | 435.0 | 53.0 | 12.0 |
| Maximum response | 568.0 | 98.0 | 26.8 |
| Maximum response/minimum response ratio | 1.31 | 1.85 | 2.33 |

With no transformation required on the responses, the next step is to determine the type of models. The Design Expert software presents several useful statistical tables that can be used to identify the type of model to select for extensive study. An important table in statistical analysis that is useful for selecting the model type is the fit summary table. The fit summary section collects the significant statistical parameters used to select the starting point for the actual model. Tables 2–4 show fit summaries for biogas yield, %COD removed, and current density, respectively. Linear, cubic, quadratic, and two-factor interaction (2FI) models are the type of models that were investigated. In statistical analysis, the model with the highest *p*-value, F-value, predicted R$^2$, and adjusted R$^2$ is regarded as the model that will best fit the data points [21]. For all responses, the quadratic model had the highest lack-of-fit *p*-value and predicted R$^2$ values. On the other hand, the cubic model had the greatest adjusted R$^2$ values, followed by the quadratic model. Therefore, these results suggested that the best fit model was either the quadratic model or the cubic model. However, the cubic model was aliased since there are no sufficient unique design points to predict all coefficients of the model. In fact, if this model was chosen, the least-squares estimations would not be unique, resulting in 2D plots with shapes that are misleading.

For this reason, the quadratic model was selected in this investigation, and because most statistical values of this type of model were the highest.

**Table 2.** Fit summary for biogas yield.

| Source | Sequential *p*-Value | Lack of Fit *p*-Value | Adjusted $R^2$ | Predicted $R^2$ | |
|---|---|---|---|---|---|
| Linear | 0.8673 | 0.0001 | −0.1948 | −0.2761 | |
| 2FI | 0.9943 | <0.0001 | −0.6277 | −0.8655 | |
| Quadratic | <0.001 | 0.0575 | 0.9982 | 0.9903 | Suggested |
| Cubic | 0.0575 | | 0.9998 | | Aliased |

**Table 3.** Fit summary for %COD removed.

| Source | Sequential *p*-Value | Lack of Fit *p*-Value | Adjusted $R^2$ | Predicted $R^2$ | |
|---|---|---|---|---|---|
| Linear | 0.5270 | 0.0011 | −0.0483 | −0.1460 | |
| 2FI | 0.9671 | 0.0008 | −0.3977 | −0.6548 | |
| Quadratic | <0.0001 | 0.0867 | 0.9904 | 0.9480 | Suggested |
| Cubic | 0.0867 | | 0.9986 | | Aliased |

**Table 4.** Fit summary for current density.

| Source | Sequential *p*-Value | Lack of Fit *p*-Value | Adjusted $R^2$ | Predicted $R^2$ | |
|---|---|---|---|---|---|
| Linear | 0.7733 | 0.0058 | −0.1549 | −0.2398 | |
| 2FI | 0.9881 | 0.0039 | −0.5639 | −8013 | |
| Quadratic | <0.001 | 0.5048 | 0.9912 | 0.9658 | Suggested |
| Cubic | 0.5048 | | 0.9918 | | Aliased |

*2.3. Analysis of Variance (ANOVA) Test and Fit Statistics*

The analysis of variance (ANOVA) test is one of the most significant tests in statistical analysis. The ANOVA test helps a researcher to observe the selected effects and the coefficients of the model [20]. Tables 5–7 depict the ANOVA outputs for biogas yield, %COD removed, and current density, respectively. F-value, Probability > F, coefficient of determination $R^2$, and lack-of-fit are statistical values that assess how the chosen regression model best fits the investigational data points [16]. An F test is used to ascertain the significance of the means between operating conditions. The F-values of 886.81, 161.67, and 175.87 denoted that the regression models were significant for biogas yield, %COD removed, and current density. The model of biogas yield had the highest F-value of 886.81, which denoted that it was the most robust model. The *p*-value is another important statistical parameter that is strongly connected with the F-value. In statistical analysis, a *p*-value denotes the probability for the regression model [20]. From the ANOVA tables, the *p*-values were less than 0.0001 for biogas yield, %COD removed, and current density. Therefore, for the biogas yield, %COD removed, and current density models, there is <0.0001% probability that F-values this large (886.81, 161.67, and 175.87) are likely due to noise. The null hypothesis has to be discarded if the *p*-value is significantly small; in other words, a *p*-value below the alpha value ($\propto$) [21,22]. This investigation was taken at a confidence interval (%CI) of 95%, thus, the alpha value was $\propto = 100 - \%CI = 5\%$. Therefore, a *p*-value below 0.05 is regarded as significant, whereas a *p*-value greater than 0.1 is not significant and thus should be ignored. For biogas yield and %COD removed, the *p*-values for the terms A, B, C, AC, $A^2$ $B^2$, and $C^2$ were below 0.05, suggesting that these model terms will have a significant effect if added to the models. On the other hand, the values that were below 0.05 for current density were A, C, AC, $A^2$ $B^2$, and $C^2$. For biogas yield and %COD removed, the terms with probability values above 0.10 were AB and BC, whereas for current density, the terms were B, AB and BC. These terms were not significant and therefore had to be removed from the models.

**Table 5.** ANOVA table for biogas yield.

| Source Model | Sum of Squares | Degree of Freedom | Mean Square | F-Value | *p*-Value Prob > F | Significance |
|---|---|---|---|---|---|---|
| Model | 27,470.32 | 9 | 3052.26 | 886.81 | <0.0001 | significant |
| A-Temperature | 1267.56 | 1 | 1267.56 | 368.28 | <0.0001 | |
| B-Voltage supply | 88.20 | 1 | 83.20 | 24.17 | 0.0044 | |
| C- Dosage | 331.53 | 1 | 331.53 | 96.32 | 0.0002 | |
| AB | 7.56 | 1 | 7.56 | 2.20 | 0.1984 | |
| AC | 228.01 | 1 | 228.01 | 66.25 | 0.0005 | |
| BC | 3.06 | 1 | 3.06 | 0.8898 | 0.3888 | |
| $A^2$ | 10,044.89 | 1 | 10,044.89 | 2918.47 | <0.0001 | |
| $B^2$ | 8064.02 | 1 | 8064.02 | 2342.94 | <0.0001 | |
| $C^2$ | 11,335.69 | 1 | 11,335.69 | 3293.50 | <0.0001 | |
| Residual | 17.21 | 5 | 3.44 | | | |
| Lack of fit | 16.54 | 3 | 5.51 | 16.54 | 0.0575 | insignificant |
| Pure error | 0.6667 | 2 | 0.3333 | | | |
| Corrected total | 27,487.53 | 14 | | | | |

**Table 6.** ANOVA table for %COD removed.

| Source Model | Sum of Squares | Degree of Freedom | Mean Square | F-Value | *p*-Value Prob > F | Significance |
|---|---|---|---|---|---|---|
| Model | 2677.94 | 9 | 297.55 | 161.67 | <0.0001 | significant |
| A-Temperature | 282.03 | 1 | 282.03 | 153.24 | <0.0001 | |
| B-Voltage supply | 58.86 | 1 | 58.86 | 31.98 | 0.0024 | |
| C- Dosage | 132.84 | 1 | 132.84 | 72.18 | 0.0004 | |
| AB | 3.24 | 1 | 3.24 | 1.76 | 0.2419 | |
| AC | 61.62 | 1 | 61.62 | 33.48 | 0.0022 | |
| BC | 2.40 | 1 | 2.40 | 1.31 | 0.3050 | |
| $A^2$ | 878.51 | 1 | 878.51 | 477.32 | <0.0001 | |
| $B^2$ | 533.91 | 1 | 533.91 | 290.09 | <0.0001 | |
| $C^2$ | 1042.12 | 1 | 1042.12 | 566.21 | <0.0001 | |
| Residual | 9.20 | 5 | 1.84 | | | |
| Lack of fit | 8.66 | 3 | 2.89 | 10.69 | 0.0867 | insignificant |
| Pure error | 0.5400 | 2 | 0.2700 | | | |
| Corrected total | 2687.14 | 14 | | | | |

**Table 7.** ANOVA table for current density.

| Source Model | Sum of Squares | Degree of Freedom | Mean Square | F-Value | *p*-Value Prob > F | Significance |
|---|---|---|---|---|---|---|
| Model | 293.35 | 9 | 32.59 | 175.87 | <0.0001 | significant |
| A-Temperature | 17.40 | 1 | 17.40 | 93.91 | 0.0002 | |
| B-Voltage supply | 0.6050 | 1 | 0.6050 | 3.26 | 0.1306 | |
| C- Dosage | 9.25 | 1 | 9.25 | 49.88 | 0.0009 | |
| AB | 0.1225 | 1 | 0.1225 | 0.6610 | 0.4532 | |
| AC | 3.80 | 1 | 3.80 | 20.52 | 0.0062 | |
| BC | 0.1225 | 1 | 0.1225 | 0.6610 | 0.4532 | |
| $A^2$ | 112.54 | 1 | 112.54 | 607.23 | <0.0001 | |
| $B^2$ | 78.84 | 1 | 78.84 | 425.39 | <0.0001 | |
| $C^2$ | 110.51 | 1 | 110.51 | 596.28 | <0.0001 | |
| Residual | 0.9267 | 5 | 0.1853 | | | |
| Lack of fit | 0.5800 | 3 | 0.1933 | 1.12 | 0.5048 | insignificant |
| Pure error | 0.3467 | 2 | 0.1733 | | | |
| Corrected total | 294.28 | 14 | | | | |

Another statistical term that is important in ANOVA is the lack-of-fit test. The selected model should have insignificant lack-of-fit [20]. This condition occurs when the *p*-value is greater than 0.10. For all models, the lack-of-fit values were above 0.10, which means the proposed regression models fit well.

Equations (1)–(3) show model equations (in coded form) for biogas yield (mL/g VS$_{fed}$), COD removed (%), and current density (mA/m$^2$), respectively [23]:

$$\text{Biogas yield} = 567.67 + 12.59 \times A - 3.23 \times B - 6.44 \times C + 7.55 \times AC$$
$$-52.16 \times A^2 - 46.73 \times B^2 - 55.41 \times C^2 \tag{1}$$

$$\text{COD removed} = 97.70 + 5.94 \times A - 2.71 \times B - 4.08 \times C + 3.92 \times AC$$
$$-15.42 \times A^2 - 12.02 \times B^2 - 16.8 \times C^2 \tag{2}$$

$$\text{Current density} = 26.47 + 1.47 \times A - 1.08 \times C + 0.9750 \times AC - 5.52 \times A^2$$
$$-4.62 \times B^2 - 5.47 \times C^2 \tag{3}$$

The model equations, expressed in terms of actual input variables, for biogas yield (mL/g VS$_{fed}$), COD removed (%), and current density (mA/m$^2$) are shown in Equations (4)–(6), respectively:

$$\text{Biogas yield} = -1531.06 + 126.29 \times (\text{temperature}) + 307.0 \times (\text{voltage supply})$$
$$+182.9 \times (\text{magnetite dosage}) + 3.36 \times (\text{temperature}) \times (\text{magnetite dosage})$$
$$-2.09 \times (\text{temperature})^2 - 186.9 \times (\text{voltage supply})^2 - 273.62 \times (\text{magnetite dosage})^2 \tag{4}$$

$$\text{\%COD removed} = -518.13 + 37.54 \times (\text{temperature}) + 80.44 \times (\text{voltage supply})$$
$$+27.11 \times (\text{magnetite dosage}) + 1.74 \times (\text{temperature}) \times (\text{magnetite dosage})$$
$$-0.617 \times (\text{temperature})^2 - 48.1 \times (\text{voltage supply})^2 - 82.96 \times (\text{magnetite dosage})^2 \tag{5}$$

$$\text{Current density} = -193.57 + 13.36 \times (\text{temperature}) + 13.7 \times (\text{voltage supply})$$
$$+0.43 \times (\text{temperature}) \times (\text{magnetite dosage}) - 0.22 \times (\text{temperature})^2$$
$$-18.48 \times (\text{voltage supply})^2 - 27.02 \times (\text{magnetite dosage})^2 \tag{6}$$

Another significant table in statistical analysis is the fit statistics table (Table 8). The table contains important statistical terms, namely standard deviation, mean, coefficient of variation, coefficient of determination ($R^2$), adjusted $R^2$, predicted $R^2$, adequate precision, and PRESS. In statistical analysis, the $R^2$ term is used to calculate how well the proposed regression model fit the investigational data points [24]. Essentially, the $R^2$ coefficient measures the percentage of change of the response variable (y) in the neighborhood of y that is described by the regression model. The coefficient of determination lies between 0 and 1. A value that is approximately equal to 1 is recommended, as it denotes that the regression model is the best fit. The model of the biogas yield was more robust than that of %COD removed and that of maximum current density, revealing a significantly high $R^2$ of 0.9994.

**Table 8.** Fit statistics for biogas yield, %COD removed, and maximum current density.

| Statistical Parameter | Biogas Yield | COD Removed | Current Density |
|---|---|---|---|
| Standard deviation | 1.86 | 1.36 | 0.43 |
| Mean | 485.37 | 74.10 | 18.14 |
| Coefficient of variation (%) | 0.3822 | 1.83 | 2.37 |
| $R^2$ | 0.9994 | 0.9966 | 0.9969 |
| Adjusted $R^2$ | 0.9982 | 0.9904 | 0.9912 |
| Predicted $R^2$ | 0.9903 | 0.9480 | 0.9658 |
| Adequate precision | 88.555 | 41.674 | 41.299 |
| PRESS | 2.18 | 1.82 | 7.80 |

However, one of the disadvantages of an $R^2$ value is that, even if the process variable is substantial, the addition of a process variable to the regression model always increases the value of $R^2$. Thus, most statisticians prefer the adjusted $R^2$ over the traditional $R^2$ [24]. One of the most important advantages of the adjusted $R^2$ is that it is not increased by the addition of a process variable. Similarly to the $R^2$, the adjusted $R^2$ of biogas was the greatest (0.9982), further proving that it is the most robust model.

The predicted $R^2$ is another important parameter in fit statistics, which indicates the estimated coefficient of determination for the proposed regression model. The predicted $R^2$ values for biogas yield, %COD removed, and current density were 0.9903, 0.9480, and 0.9658, while the adjusted $R^2$ values were 0.9982, 0.9904, and 0.9912, respectively. The respective difference between the predicted $R^2$ and adjusted $R^2$ were 0.0079, 0.0424, and 0.0432. The differences were less than 0.2, which suggested reasonable agreement. In other words, there was no problem with either the experimental data or the regression models.

The statistical term adequate precision evaluates the limits of the estimated output to the predicted error; in other words, the ratio of signal/response-to-noise [21]. A high adequate precision denotes an extremely high difference between the estimated response output and the accompanying error. Generally, an adequate precision that is above 4.0 denotes adequate model discrimination [21]. In this investigation, the adequate precisions for biogas yield, %COD removed, and current density were 88.55, 41.67, and 41.29, respectively. These values were above 4.0, which indicated that the model discrimination was satisfactory. Therefore, the selected model expressions can be used to navigate the design spaces of the responses because the predicated response outputs are less influenced by error. The highest adequate precision of 88.55 was found on the model of biogas yield. This means that this model had the highest signal to noise ratio, which further proves that it is the most robust model.

The coefficient of variation denotes the standard deviation represented as a percentage of the mean of the response variable [20]. Essentially, the coefficient of variation is used to measure the dispersion of experimental data points around the mean. A low value is recommended since it indicates a more authentic model equation. The coefficient of variation of the biogas yield, %COD removed and current density models were 0.3822%, 1.83%, and 2.37%, respectively. These values were low, with the biogas yield revealing the lowest value of 0.3822%, thus indicating a more robust model. Thus, the standard deviation of biogas yield (1.86) was extremely low compared to its associated mean of 485.37, which denotes that its regression model was more satisfactory than the other models.

Predicted residual error sum of squares (PRESS) is used to calculate the error variation. Specifically, this statistical coefficient illustrates how the variation in the process variable in a model equation cannot be described by the regression model [20]. Generally, a very low PRESS indicates that the regression model is the best fit. On the other hand, a high PRESS is an indication that the model equation is not the best fit. The PRESS values for biogas yield, %COD removed, and current density were 2.18, 1.82, and 0.7.80, respectively. These values were low, which suggested that each of the models was a best fit.

### 2.4. Validation of the Models

Diagnostic tests are extremely important in statistics to confirm that the assumptions for the ANOVA test are met; the regression models must be validated [21]. The predicted versus actual plot (Figure 2) is one of the useful plots for model validation. The response values of the predicted versus actual plot should be scattered along the 45° line [21]. As is evident from the figures, the points of biogas yield (Figure 2a), %COD removal (Figure 2b), and current density (Figure 2c) were scattered along the 45° line, indicating that all regression models were able to reasonably predict the experimental data points.

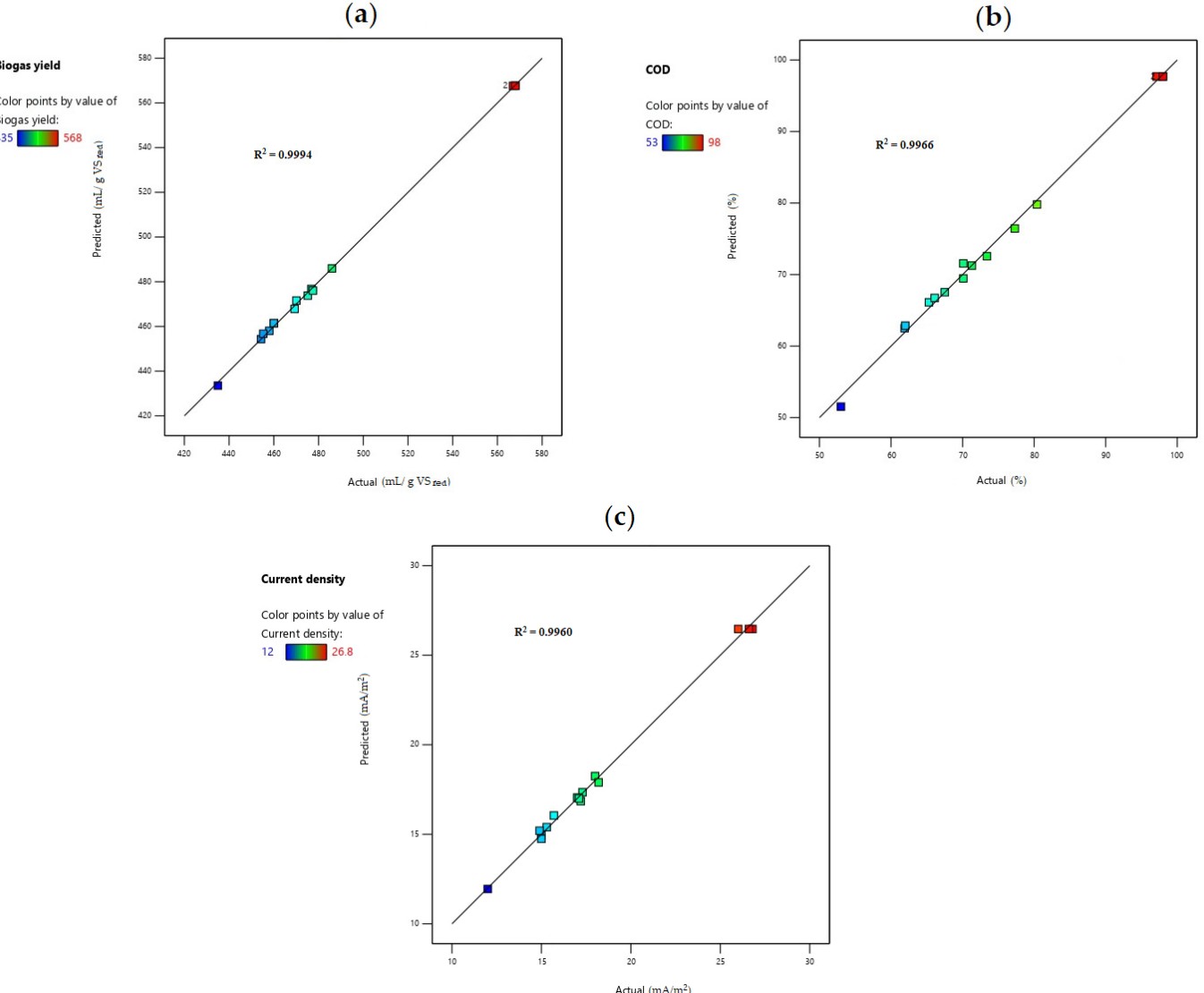

**Figure 2.** Predicted versus actual values of (**a**) biogas yield; (**b**) %COD removed; and (**c**) current density.

The leverage plot is another useful graph in model validation (Figure 3). The average leverage ($L_{average}$) may be defined by Equation (7):

$$L_{average} = N_{mc}/N_{er} \qquad (7)$$

where $N_{mc}$ denotes the number of model coefficients, and $N_{er}$ is the total number of experimental runs.

In this investigation, the value of $N_{cm}$ was 9 and that of $N_{er}$ was 15; therefore, according to Equation (7), the average leverage was 0.6. An experimental run with a leverage value greater than two times $L_{average}$ or a leverage value of 1 is regarded as having a very high value of leverage [21]. From the biogas yield (Figure 3a), %COD removed (Figure 3b), and current density (Figure 3c) graphs, there is no leverage value above 1 or greater than 2 × 0.6 = 1.2, denoting that the leverage values were not too high.

The last diagnostic tool that was used to validate the models was the externally studentized residuals versus run number graph (Figure 4). The externally studentized residuals versus run number plot portrays the number of t-values in an investigational run that fall off from the rest of the data points; in other words, "Outliers" [20]. The graph

has boundary lines that are dependent on the degree of freedom as well as the tail value. Equation (8) may be used to calculate the tail value:

$$\text{tail value} = \propto / (2N_{er}) \tag{8}$$

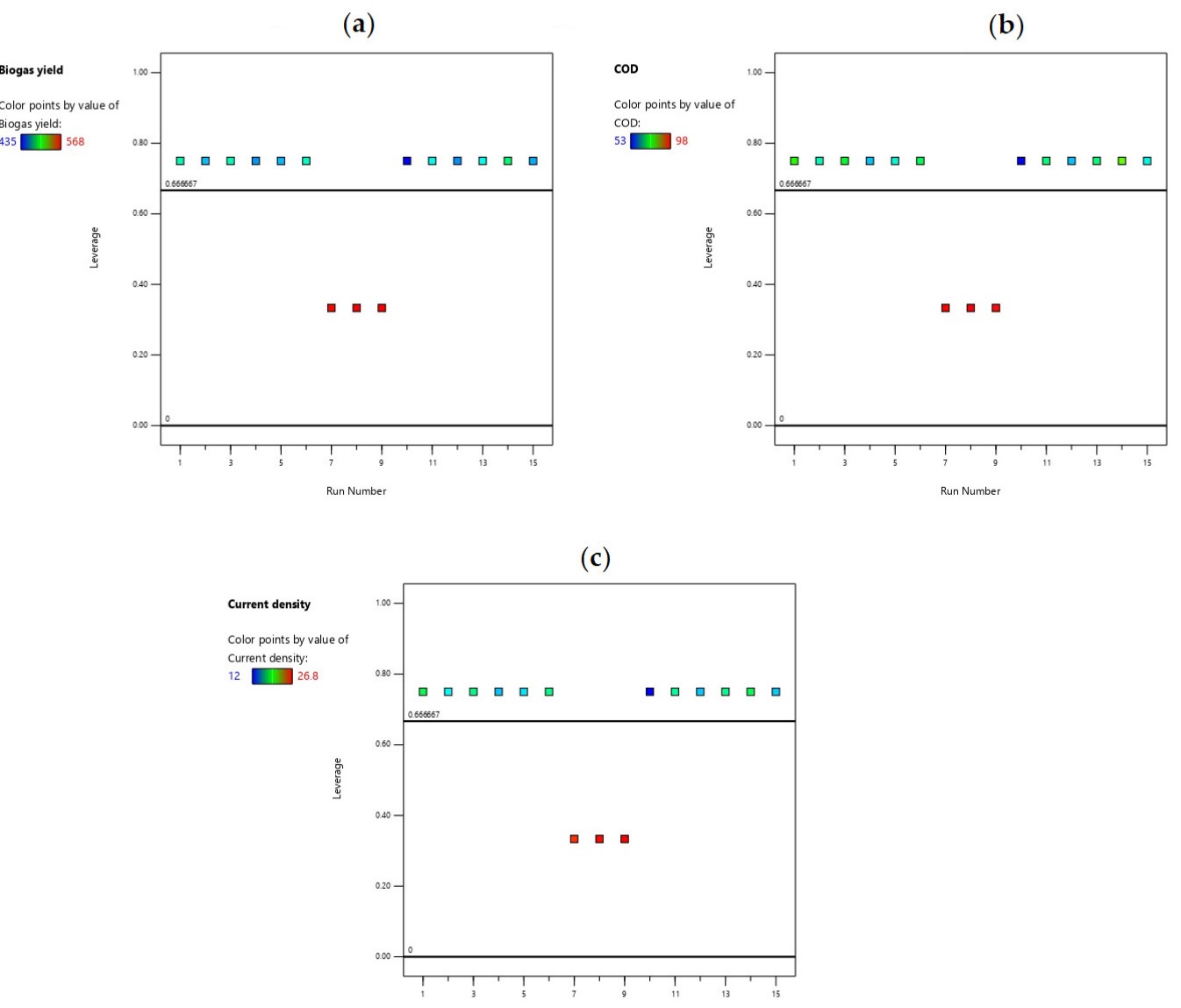

**Figure 3.** Leverage versus run for (**a**) biogas yield; (**b**) %COD removal; and (**c**) current density.

The alpha value ($\propto$) is 0.05 while the total number of experimental runs ($N_{er}$) is 15. Therefore, according to Equation (8), the tail value is 0.0017.

The degree of freedom for residual (DOF) may be obtained from Equation (9):

$$\text{DOF} = N_{er} - n - 1 \tag{9}$$

where n is the number of process variables.

In this investigation, the value of n was 3. Therefore, according to Equation (9), the degree of freedom for residual is 5.

According to the one-tailed/two-tailed t-table [21], at a degree of freedom of 5 and tail value of 0.0017, the residual limit ($L_r$) is $L_r = \pm t_{(\text{tail value,DOF})} = \pm t_{(0.0017, 5)} = \pm 6.25$.

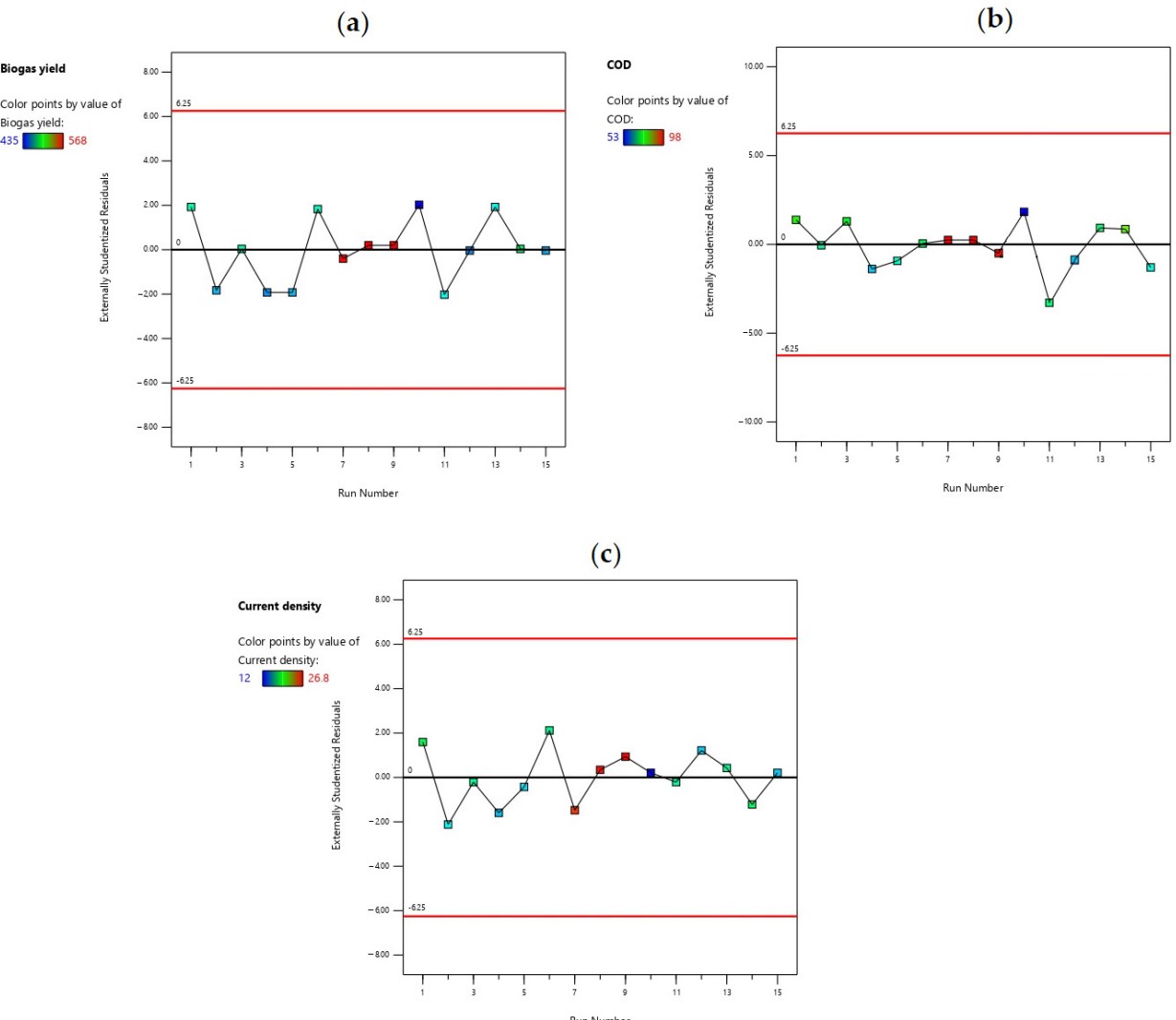

**Figure 4.** Residuals versus experimental runs for (**a**) biogas yield; (**b**) %COD removed; and (**c**) current density.

From the graphs of biogas yield (Figure 4a), %COD removed (Figure 4b), and current density (Figure 4c), all externally studentized residuals were within the residual limits ($L_r$), suggesting that the responses need not be transformed. Therefore, all investigational data points fitted well by the models, which suggests that there was no problem with either the experimental data points or the regression models.

*2.5. Surface Graphs*

A three-dimensional surface (3D) plot is a projection of a contour plot. A 3D plot makes it simpler to see the actual representation of a model. The effect of the process variables on biogas yield, %COD removed, and current density is depicted in Figure 5. Figure 5a,b display 3D graphs for biogas yield of voltage supply and temperature (BA), and magnetite nanoparticle dosage and temperature (CA), respectively. The results show that biogas yield firstly increases with either an increase in temperature or voltage supply or magnetite nanoparticle dosage. Temperature is essential in anaerobic digestion since it enhances the metabolic rate, and thus biogas yield. According to Ohm's law, voltage is directly proportional to current. Therefore, increasing voltage will increase current density and electrical conductivity, and hence biogas yield. On the other hand, magnetite nanoparticles can help to enhance the interspecies electron transfer between archaea and microorganisms, which improves biogas yield. However, biogas yield will reach the maximum value of

563.0 mL/g $VS_{fed}$ when temperature, voltage supply, and magnetite nanoparticle dosage are at about 32 °C, 0.77 V, and 0.53 g, respectively, which is at a medium level. After passing the optimum conditions, biogas yield then decreased with an increase in any of the process variables. Essentially, the activity of microorganisms decreases as the temperature of the system becomes higher, which as a result slows down the microbial production of protons and electrons [25]. Therefore, biogas yield decreased above 32 °C. On the other hand, high voltage results in water hydrolysis, which leads to hydroxides and oxygen, which hinder the digestion process [26]. Therefore, biogas yield was reduced after 0.77 V. With regards to magnetite nanoparticle dosage, higher dosages generally result in toxicity that hinders the activity of the methanogenesis stage, which is why above 0.53 g, biogas yield decreased.

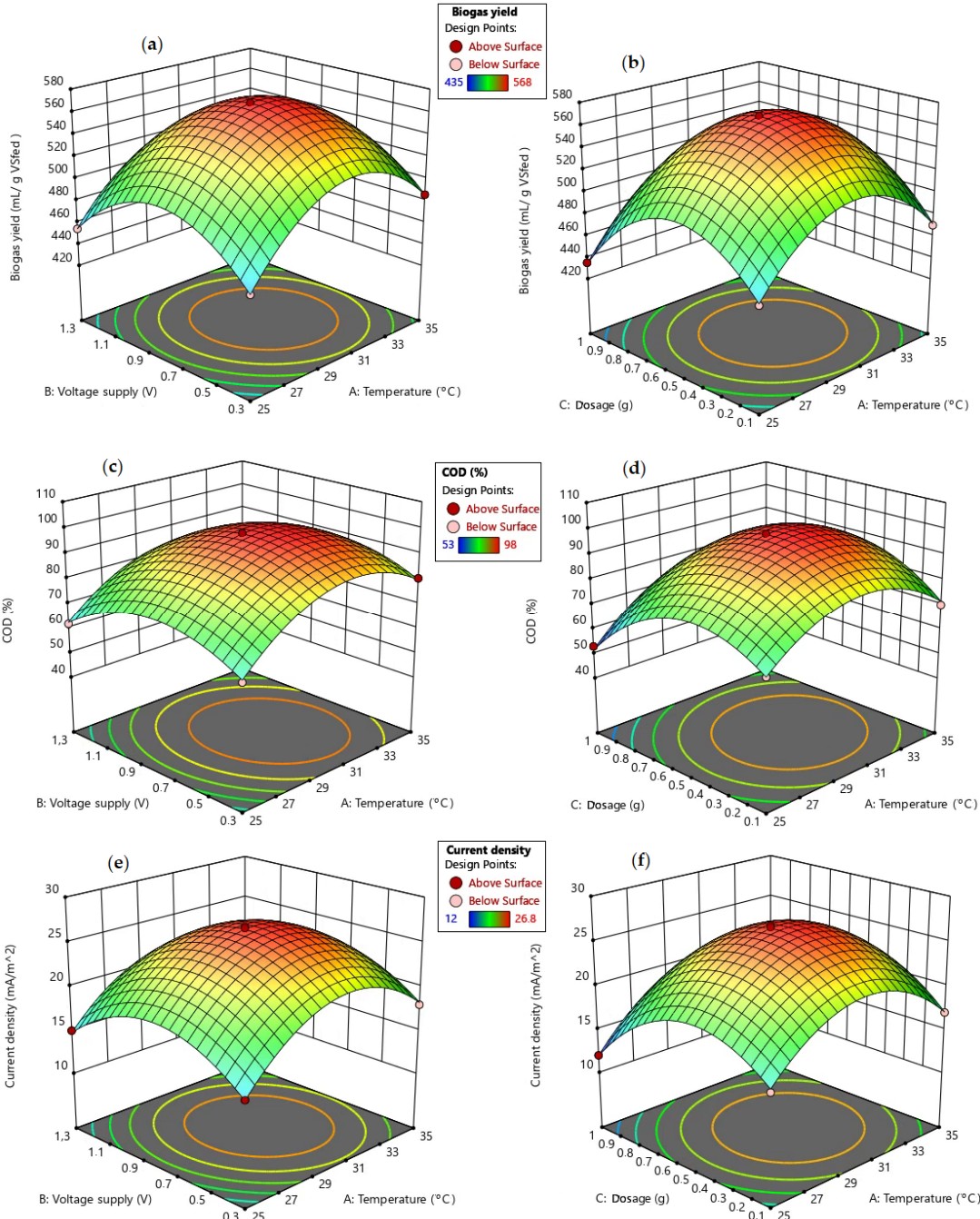

**Figure 5.** 3D graphs of BA and CA for (**a**,**b**) biogas yield; (**c**,**d**) %COD removed; and (**e**,**f**) current density, respectively.

From Figure 5c,d, the optimum %COD removed was approximately 97% and was achieved at a temperature of 32 °C, voltage supply of 0.77 V, and magnetite nanoparticle dosage of 0.53 g. Likewise, current density had the highest value of approximately 26 mA/m$^2$ (Figure 5e,f), which occurred at the same optimum operating conditions as for biogas yield and %COD removal. Most researchers have found the optimum temperature value to be 35 °C for a bioelectrochemical system [27,28], which is higher than what was obtained, hence, their systems employed more energy. According to Xu et al. [4], the optimum voltage supply is 0.8 V. The optimum value that was obtained from this investigation was closer to this value.

Even though optimum conditions have been found for biogas yield, %COD removed, and current density using a graphical method, the optimum conditions have to be confirmed by means of a numerical optimization method.

*2.6. Numerical Optimization Ramps and Desirability*

The last step in this investigation was to find the optimal solution using a numerical method. Numerical optimization ramps were used to obtain the optimum temperature, voltage supply, and magnetite nanoparticle dosage with the aim of optimizing the response variables. The possible goals were set to maximize the responses. Unlike a five-pluses method that requires one goal to be important, all goals were equally important in this investigation, and therefore, the goal importance was set to three pluses (+++) [21]. According to Figure 6, at optimal process conditions of temperature (32.2 °C), voltage supply (0.77 V), and magnetite nanoparticle dosage (0.53 g), the Design Expert software revealed that the optimum values for biogas yield, %COD removed, and maximum current density were 563.02 mL/g VS$_{fed}$, 97.52%, and 26.05 mA/m$^2$, respectively. At the optimum conditions, the results revealed a maximum combined desirability of 89.9%.

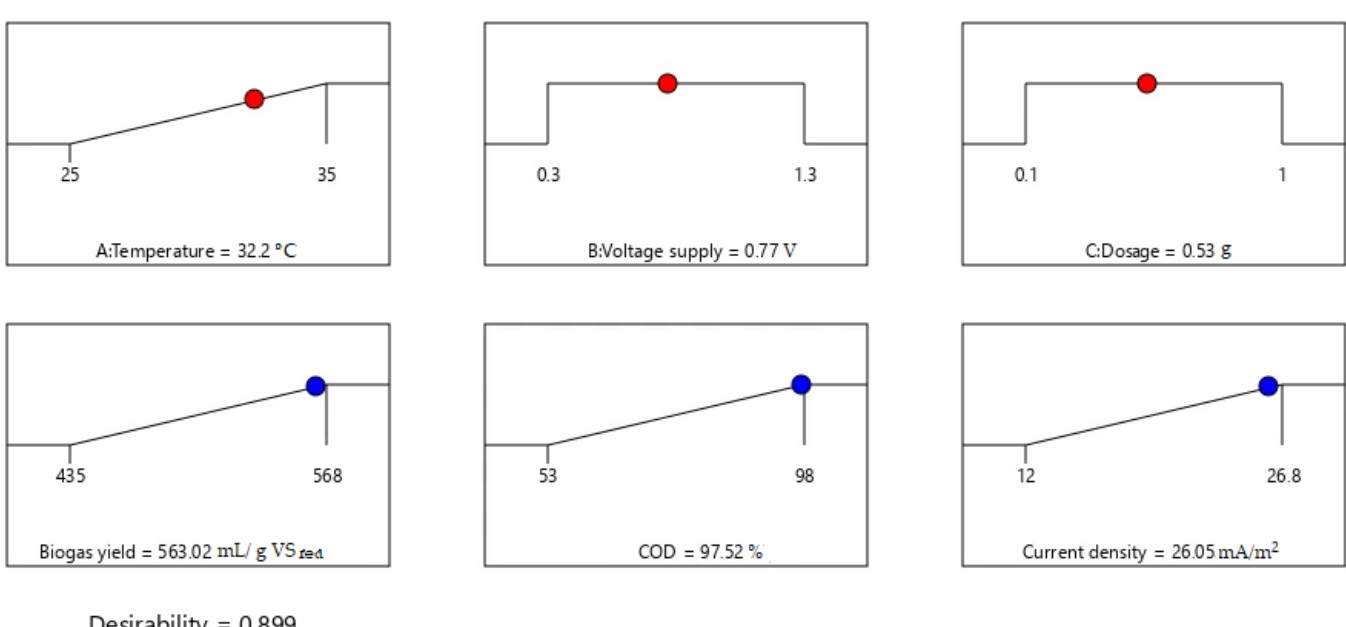

**Figure 6.** Ramp graphs of optimized process variables for the investigated responses.

Our previous studies on the synergistic effect of the bioelectrochemical system and magnetite nanoparticles were conducted at 40 °C and at a magnetite dosage of 1 g [2,5,19] as shown in Table 9. Out of these studies, the best performing bioelectrochemical system was a MEC with cylindrical electrodes, which revealed the highest biogas yield of 548.0 mL/g VS$_{fed}$ [19]. However, the current investigation had a higher optimum biogas yield of 563.02 mL/g VS$_{fed}$ (than our previous investigations), which was found at a lower

temperature of 32.2 °C and lower magnetite dosage of 0.53 g. Thus, the current study has improved the performance of the bioelectrochemical system while reducing energy usage and magnetite consumption.

**Table 9.** Comparison of previous studies and this study.

| Parameter | Unit | Study [2] | Study [5] | Study [19] | This Study |
|---|---|---|---|---|---|
| Temperature | °C | 40.0 | 40.0 | 40.0 | 32.2 |
| Magnetite nanoparticle dosage | g | 1.00 | 1.00 | 1.00 | 0.53 |
| Biogas yield | mL/ g VS$_{fed}$ | 404.4 | 441.2 | 548.0 | 563.02 |

Figure 7 depicts the response surface of desirability performance in terms of A: temperature, B: voltage supply, and C: magnetite nanoparticle dosage. Figure 7a,b prove that the desirability value increased with an increase in temperature for values between 25 and 32.2 °C. This is due to the fact that, by increasing the temperature from atmospheric values to mesophilic values, the electrical conductivity of a solution is improved, which then enhances the flow of electrons and protons, and hence, biogas production and contaminants removed. Thus, the desirability is increased by an increase in temperature. However, if the temperature is above 32.2 °C, certain other factors, such as instability and low microbial activity, should be taken into consideration, as they reduce the performance of a MFC [29].

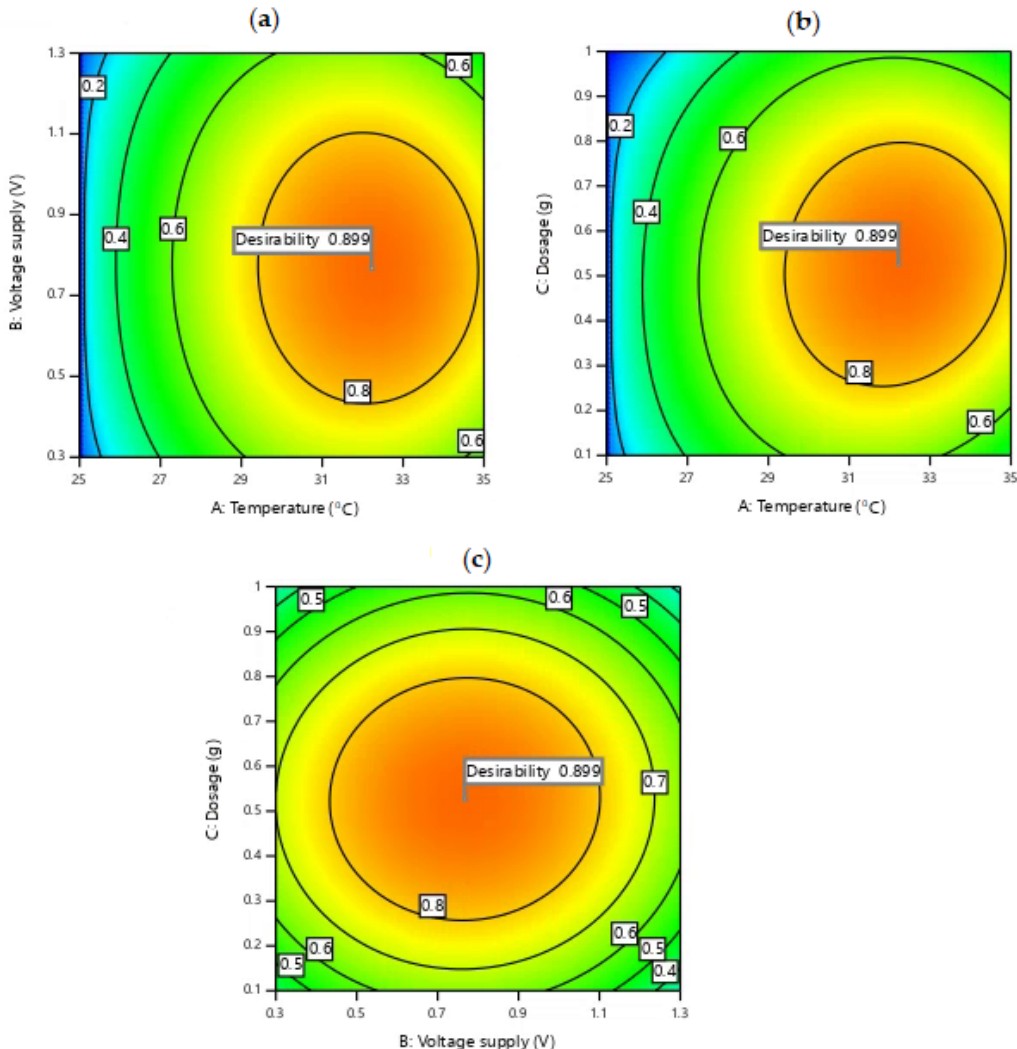

**Figure 7.** Contour plots depicting the response surface of desirability performance in terms of operating conditions: (**a**) BA; (**b**) CA; and (**c**) CB.

On the other hand, the vertical axis of Figure 7a and the horizontal axis of Figure 7c shows desirability as a function of voltage supply. It is evident from the graphs that, in general, if the amount of voltage supply is increased for values between 0.3 and 0.77 V, the desirability is increased. However, the graph approaches the optimum value of 0.77 V and undergoes a transformation; the desirability performance begins to drop with an increase in voltage supply. Generally, pH divergence is evident at a high voltage supply and the activity of the bioanode is usually inhibited due to the formation of toxic substances, thus reducing the overall performance of a MEC [26].

The *y*-axis of Figure 7b,c depicts the desirability of the system for maximum magnetite nanoparticle dosage. As is evident from the graphs, for magnetite nanoparticles dosage between 0.1 and 0.50 g, an increase in dosage results in an increase in desirability. The figures confirm that after 0.50–0.53 g magnetite nanoparticle dosage, desirability begins to decrease, possible due to inhibition.

Experimental work was then executed at the optimum process conditions (32.2 °C, 0.77 V, and 0.53 g) with the aim of validating the results of the predicted models. A deviation of below 4% was obtained when comparing the two results, which indicated that the results of the models were more or less the same as the experimental data. Therefore, the estimated model equations can be used for the synergism of a microbial electrolysis cell and magnetite nanoparticles at any combinations of temperature, voltage supply, and magnetite nanoparticle dosage that lie within the investigated ranges.

### 3. Materials and Methods

#### 3.1. Microbial Electrolysis Cells Setup and Operation

Duran–Schott bottles (15 digesters) were used to carry out the experimental work, with a total working volume of 800 mL. Figure 8 shows the equipment setup for the MEC digesters. The feed to each digester consisted of 300 mL of inoculum (sewage sludge), 500 mL of substrate (waste-activated sludge) and magnetite nanoparticles. To flush out air in the digesters, the headspace of each reactor was firstly supplied with nitrogen gas (99.9%). The experimental work was executed for 30 days, and the temperature of the digesters was maintained by a circulating water bath. The top cap of each digester had four ports for transferring gas to the water displacement system, sampling, and anode and cathode electrodes. The anodic electrode (zinc) and cathodic electrode (copper) were 5 cm apart to minimise ohmic resistance and were both connected to a DC power supply (Matrix MPS3005S, Shenzhen, China). Each electrode had a width of 1 cm and a length of 12 cm.

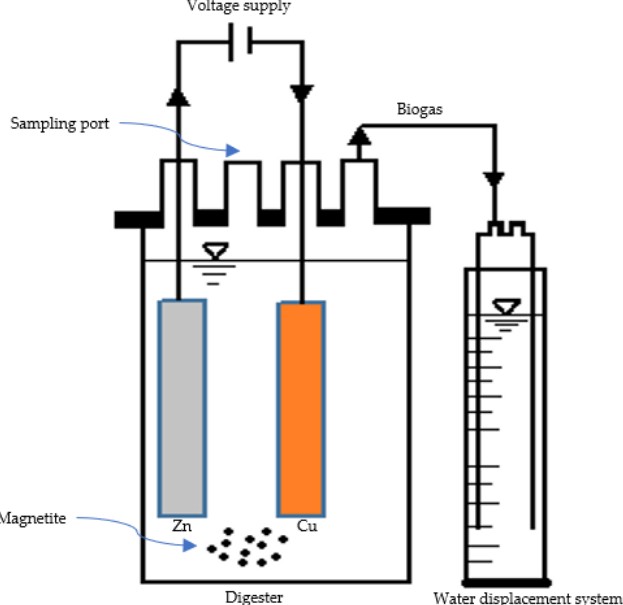

**Figure 8.** Equipment setup for the MEC digesters.

### 3.2. Magnetite Nanoparticles Synthesis and Reagents

The magnetite nanoparticles used in this study were obtained from the magnetite nanoparticles synthesized by Amo-Duodu et al. [30]. A co-precipitation technique was employed for the synthesis of magnetite nanoparticles. The magnetite nanoparticle synthesis involved the adding of chemical reagents, ferrous sulfate heptahydrate, nickel (II) nitrate hexahydrate, oleic acid, ferric chloride hexahydrate, and sodium hydroxide. The 1 M sodium hydroxide and ferrous sulfate heptahydrate (purity > 99%) were bought from Labcare Supplies (PTY) LTD, whereas ferric chloride hexahydrate (purity > 99%) was purchased from United Scientific SA cc, South Africa. Nickel (II) nitrate hexahydrate and oleic acid (purity > 99%) were obtained from Sigma–Aldrich, South Africa. The X-ray diffraction confirmed the face-centered cubic shape structure and a crystal size of 5.179 nm [30].

Both the substrate and inoculum were obtained from a Durban-based wastewater treatment industry, in the province of KwaZulu-Natal, South Africa. The substrate and inoculum were sampled by means of 20 L containers. The samples were homogenized and kept at atmospheric conditions before being used.

### 3.3. Chemical Analyses and Electrochemical Characteristics Measurement

Daily biogas generation was measured using a water displacement system. Biogas comprised mostly of 60% methane and 40% carbon dioxide [31]. Therefore, only methane and carbon dioxide were taken into consideration when compensating for gas dissolved in water. Methane is less soluble in water (for example, 0.017 g methane/kg of water at 1 atm and 35 °C), whereas carbon dioxide is extremely soluble in water (1.20 g carbon dioxide/kg of water at 35 °C and 1 atm). Nevertheless, the amount of gas that was dissolved in the contents of the volumetric system was taken into account using the solubility values from Perry et al. [32].

One of the most significant anaerobic parameters, chemical oxygen demand (COD), was measured before and after the experimental work by means of COD vials together with a Hach DR 3900 spectrometer (Hach Company, Loveland, CO, USA). Standard methods for the examination of water and wastewater as proposed by the American Public Health Association (APHA) were used for wastewater characterization [33]. The amount of %COD removed ($COD_{removed}$) was determined by Equation (10):

$$\% \, COD_{removed} = \frac{COD_{feed} - COD_{effluent}}{COD_{feed}} \times 100\% \tag{10}$$

where $COD_{feed}$ is the COD of the feed (mg/L), and $COD_{effluent}$ is the COD of the effluent (mg/L).

The pH was measured by means of a Hanna H198129 conductivity meter. The TSS and color were measured by a Hach DR 3900 spectrometer (Hach Company, Loveland, CO, USA)

The physiochemical characteristics of the influent wastewater are shown in Table 10:

**Table 10.** The physiochemical characteristics of the influent wastewater used in this study.

| Parameters | Unit | Amount |
|---|---|---|
| pH | - | 6.43 ± 0.33 |
| Total suspended solids | mg/L | 39.35 ± 1.44 |
| Chemical oxygen demand (COD) | mg/L | 2298.57 ± 200.37 |
| Color | Pt.Co | 253.48 ± 5.32 |

The current flow and cell voltage were measured by means of a FLUKE 177 RMS multimeter (FLUKE Company, Washington, USA). Current density (j) was calculated using Equation (11):

$$j = \frac{I}{A} \tag{11}$$

where I represents current (mA), and A is the area of the electrode ($m^2$).



*3.4. RSM*

A Box–Behnken design method of experiments for optimization and modelling, employing RSM and encompassing three input variables (factors), A: temperature, B: voltage supply, and C: magnetite nanoparticle dosage, was implemented as the RSM. The responses that were measured were biogas yield, %COD removed and current density. According to Li et al. [34], the optimum temperature for a microbial electrolysis cell and microbial fuel cell is in the range of 25 and 35 °C. Therefore, the experimental work was investigated at 25, 30 and 35 °C. The best-performing microbial electrolysis cell has been found to be within the voltage supply range of 0.3 and 1.0 V [4,26,35]. In this investigation, the voltage supply was tested at 0.3, 0.8 and 1.3 V. The main reason for going above 1.0 V was to investigate whether the microbial electrolysis cell would be inhibited or not. The anaerobic digester is typically inhibited if the magnetite nanoparticle dosage is more than 0.4 g [36]; therefore, the magnetite nanoparticle dosage was tested at 0.1, 0.55 and 1.0 g.

The number of experimental runs ($N_{er}$) in a Box–Behnken design method is determined by Equation (12) [13].

$$N_{er} = n_{fp} + P_{cp} = 2n_{iv}(n_{iv} - 1) + P_{cp} \tag{12}$$

where $n_{fp}$ is factorial points, $P_{cp}$ represents the number of centre points, and $n_{iv}$ denotes the number of input variables. In this study, the number of input variables ($n_{iv}$) was 3 and the number of centre points ($P_{cp}$) was 3. Therefore, the number of experimental runs ($N_{er}$), according to Equation (12), was 15. The design matrix for the 15 experimental runs is shown in Table 11.

**Table 11.** Design matrix for the Box–Behnken design method.

| | Actual Input Variables | | | Coded Input Variables | | |
|---|---|---|---|---|---|---|
| Run | A: Temperature (°C) | B: Voltage Supply (V) | C: Dosage (g) | A | B | C |
| 1 | 30 | 0.3 | 0.10 | 0 | −1 | −1 |
| 2 | 25 | 0.8 | 0.10 | −1 | 0 | −1 |
| 3 | 35 | 1.3 | 0.55 | 1 | 1 | 0 |
| 4 | 30 | 1.3 | 1.00 | 0 | 1 | 1 |
| 5 | 30 | 0.3 | 1.00 | 0 | −1 | 1 |
| 6 | 35 | 0.8 | 1.00 | 1 | 0 | 1 |
| 7 | 30 | 0.8 | 0.55 | 0 | 0 | 0 |
| 8 | 30 | 0.8 | 0.55 | 0 | 0 | 0 |
| 9 | 30 | 0.8 | 0.55 | 0 | 0 | 0 |
| 10 | 25 | 0.8 | 1.00 | −1 | 0 | 1 |
| 11 | 35 | 0.8 | 0.10 | 1 | 0 | −1 |
| 12 | 25 | 1.3 | 0.55 | −1 | 1 | 0 |
| 13 | 30 | 1.3 | 0.10 | 0 | 1 | −1 |
| 14 | 35 | 0.3 | 0.55 | 1 | −1 | 0 |
| 15 | 25 | 0.3 | 0.55 | −1 | −1 | 0 |

A stepwise regression was used to examine the experimental data and a second-order regression model was employed for the best fit so as to identify the relevant terms of the mathematical model. The second-order polynomial, such as the Box–Behnken design, can be approximated by the expression in Equation (13) [23]:

$$y_r = \beta_0 + \sum_{l=1}^{p} \beta_l x_l + \sum_{r=1}^{N_{er}} \beta_{rr} x_r^2 + \sum \sum_{l>r} \beta_{rl} x_r x_l + \varepsilon \tag{13}$$

where $\beta_0$ denotes the regression coefficient at $r = 0$; $\beta_l$ is the $l^{\text{th}}$ regression coefficient; $l = 1, 2, 3, \ldots, p$; $x_l$ is the $l^{\text{th}}$ input variable; $\beta_{rr} x_r^2$ denotes quadratic expressions; $\beta_{rl} x_r x_l$ represents cross product expressions; and the term $\varepsilon$ is the experimental error.

The software that was used for optimization and modelling was Design Expert 12 V.12.0.0 (Stat-Ease Incorporated, Minneapolis, MN, USA). Design Expert allows data fitting of models with coefficients that are not known. Lack-of-fit and sequential F-test, together with adequacy measures, were then executed in finding the best regression equations.

## 4. Conclusions

The focus of this paper was on the utilization of RSM on the synergism of MECs and magnetite nanoparticles for wastewater treatment using the Box–Behnken design method. The end results of biogas yield, %COD removed, and maximum current density indicated that the best-fit model was a quadratic model, which demonstrated the greatest *p*-value, predicted $R^2$, and adjusted $R^2$ values. The model of the biogas yield was more robust than that of %COD removed and maximum current density models, revealing a significantly high $R^2$ of 0.9982. Second-order regression models were also generated using Design Expert software to predict biogas yield, %COD removed, and maximum current density at the combinations of temperature, voltage supply, and magnetite nanoparticle dosage. The models were validated by predicted versus actual plots, residuals versus runs, and leverage versus run number graphs. The outcome of the results revealed that the optimal MEC conditions were: voltage supply of 0.77 V, temperature of 32.2 °C, and magnetite nanoparticle dosage of 0.53 g. Under these optimum conditions, the RSM revealed the highest biogas yield of 563.02 mL/g $VS_{fed}$, %COD removed of 97.52%, and maximum current density of 26.05 mA/m$^2$. The results revealed a desirability performance of 89.9% for biogas yield, COD removed, and maximum current density. Furthermore, the influence of the operating conditions on the responses followed the order temperature > voltage supply > magnetite nanoparticle dosage. Therefore, the synergism of MECs and magnetite nanoparticles is mostly governed by temperature. In conclusion, this study showed the effective utilization of statistical modeling and optimization to enhance the performance of the MEC to achieve a sustainable and eco-friendly situation.

**Author Contributions:** Conceptualization, N.I.M.; methodology, N.I.M.; validation, N.I.M.; investigation, N.I.M.; data curation, N.I.M.; writing—original draft preparation, N.I.M.; writing—review and editing, S.R. and B.F.B.; supervision, S.R. and B.F.B. All authors have read and agreed to the published version of the manuscript.

**Funding:** This study was sponsored by the National Research Foundation (NRF), grant number 129076, Department of Chemical Engineering at Durban University of Technology.

**Data Availability Statement:** Not applicable.

**Acknowledgments:** The authors would like to thank the Green Engineering Research Group and Durban University of Technology.

**Conflicts of Interest:** The authors declare no conflict of interest.

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
