# Peer review of "Utilization of Response Surface Methodology in Optimization and Modelling of a Microbial Electrolysis Cell for Wastewater Treatment Using Box–Behnken Design Method"

_catalysts, doi:10.3390/catal12091052_

Round 1

Reviewer 1 Report

The work presented for review focuses on utilization of RSM on the synergistic of MEC and magnetite-nanoparticles for wastewater treatment using the Box-Behnken Design method. The end results of biogas yield, %COD removed, and maximum current density indicated that the best fit model was a quadratic model, which demonstrated the greatest p-value, predicted R2, and adjusted R2 values. The model of the biogas yield was more robust than that of %COD removed, and maximum current density models, revealing a significantly high R2 of 0.9982. Second-order regression models were also generated using Design Expert software to predict biogas yield, %COD removed, and maximum current density at the combinations of temperature, voltage supply, and magnetite-nanoparticle dosage. The article presents a very extensive research proving that  the optimal MEC conditions were: voltage supply of 0.77 V, temperature of 32.2°C, and magnetite-nanoparticles dosage of 0.53 g. Under these optimum conditions, the RSM revealed the highest biogas yield of 563.02 mL/ g VSfed, %COD removed of 97.52%, and maximum current density of 26.05 mA/m2. The results revealed a desirability performance of 89.9% for biogas yield, COD removed, and maximum current density.

The article presents a large amount of literature related to the research topic.

·       However, please refine, revision, correction the drawings, e.g. figure 7 does not include units, figures 3, 4 and 5 also require.

·       The writing of figures in the text also requires standardization.

Thank you for considering my opinion. I encourage the authors to continue working on improving the manuscript.

Author Response

Please find attach file for your reference

Reviewer 2 Report

Please find my comments on your article

1- Could you draw the set up of experimental and label all parts?

2- In line 176, the unit of current is mA, double check please.

3- In line 200, to calculate Ner use Equation 3 not 1.   

4- Could you add the references of Equation 4, 5, 6, and 7.

5- In tables, could you add the units of biogas yield and current density?

Author Response

(The authors gave the same response as above.)
